# Artificial Radiation Frost Chamber for Frost Formation on Tea

**Yong-Zong Lu [1,2]** **, Yong-Guang Hu [1,\*], Jin-Tao Tian [1], Huan Song [1] and Richard L. Snyder [2]**

1    School of Agricultural Equipment Engineering, Jiangsu University, Zhenjiang 212013, China;
     luyongzong@126.com (Y.-Z.L.); 18531244229@163.com (J.-T.T.); 2111816009@stmail.ujs.edu.cn (H.S.)
2    Department of Land, Air and Water Resources, University of California, Davis, CA 95616, USA;
     rlsnyder@ucdavis.edu
\*    Correspondence: deerhu@ujs.edu.cn; Tel.: +86-138-1515-1176

**Abstract:** The Yangtze River region is the main production area for famous, high-quality tea in China. Radiation frost frequently occurs in this region, especially in the early spring during calm and clear nights, and it causes substantial damage to crops, which leads to huge economic losses for tea growers. The formation of frost is difficult to experimentally control due to the complexity and variability of the agro-micrometeorological environment. The objective of this study was to evaluate an artificial radiation frost chamber based on the temperature difference between leaf and air dew point, which was designed for advanced frost-related research. Micro-meteorological data and the frost formation process were monitored in an experimental tea field during typical radiation frost nights to mimic declining temperatures that are consistent with nature. The radiation frost chamber model and main parameters were determined by theoretical calculations and computational fluid dynamics (CFD) simulation. A frost-forming experiment was conducted to evaluate the performance of the frost chamber. The observation results showed that the greatest temperature difference between leaf and air dew point ($T_{differ}$) was −2.3 °C. The simulation results showed that the desublimation cooling rate of the air vapor was greater than sublimation, and the $T_{differ}$ should be greater than −3.2 °C, which could cause frost to easily form on the leaf. The performance testing results showed that leaf temperature slowly declined after a rapid decrease, which is similar to the natural condition, which results in noticeable frost formation on the leaf.

**Keywords:** air dew point; CFD simulation; leaf-air temperature difference; leaf temperature; radiation-frost chamber

## 1. Introduction

Frost refers to the formation of ice crystals on surfaces, either by freezing of dew or a phase change from vapor to ice, which causes injurious frost formation on crops and plants [1,2]. It occurs at temperatures of less than or equal to 0 °C, as measured in a "Stevenson-screen" shelter at a height between 1.25 and 2.0 m [3]. It can also occur with an air temperature less than 0 °C, without defining the shelter type and height [4]. Snyder et al. [2] have subdivided frost into advection frost and radiation frost. Advection frosts are associated with large-scale incursions of cold air with a well-mixed, windy atmosphere and a temperature that is often subzero, even during the daytime. Radiative frosts are associated with cooling due to energy loss through radiant exchange during clear, calm nights, and with air temperature inversions. In some cases, a combination of both advection and radiation conditions will occur.

Radiation frosts commonly occur in the Yangtze River region, China, especially during early spring nights [5,6], and they are characterized by clear skies, calm winds, and air temperature inversions [7,8].

Radiation frosts can be categorized into "hoar frost" and "black frost". A "hoar frost" occurs when water vapor deposits onto the surface and forms a white coating of ice that is commonly called "frost". A "black" frost occurs when the temperature falls below 0 °C and no ice forms on the surface. Black frost occurs when the humidity is sufficiently low. When the humidity is high, ice is more likely to deposit and a "hoar frost" can occur. Radiation frosts occur because of heat losses in the form of radiant energy [3]. Under clear night-time skies, more heat is radiated away from the surface than is received, which thus causes the air temperature to decrease [9]. The air temperature decreases faster near the radiating surface, causing the formation of a temperature inversion. The sensible heat content of the soil surface and air near the surface decreases, as there is a net loss of energy through radiation from the surface. There is a flux of sensible heat downward from the air and upward from within the soil to the surface to replace the lost sensible heat. The dew point is the temperature at which air must be cooled to become saturated with water vapor. When further cooled, the airborne water vapor will condense to form liquid water [10]. When air cools to its dew point through contact with a surface that is colder than the air, water will condense on the surface. When the air temperature is below the freezing point of water, the dew point is called the frost point, as frost is formed rather than dew point [2,11].

Recent research on radiation frost focused on the mechanism of frost formation, critical temperature measurement, and the freezing injury of radiation frost [9,12–14]. The use of artificial radiation-frost chamber is crucial to frost research, as it simulates a natural radiation frost environment. However, most of the frost studies used an incubator based on the convective heat transfer between plants and the air to decrease the environment temperature [15–17]; this is inconsistent with the mechanism of a natural radiation frost event, and thus will negatively affect the experimental accuracy.

Due to our limited ability of retrieval, we found scant research regarding the design of a frost chamber in the past decades. Marcellos (1981) [18] described a small chamber for studying plant reaction to radiation frost that incorporated a radiation sink comprising a blackened, refrigerated brass surface, and walls whose temperature can be controlled for convective exchange. The chamber produced a kind of dendritic crystalline frost on the surfaces of wheat plants. Phillips (1984) [19] designed a frost-producing device that consisted of temperature heating, refrigeration, water vapor spraying, and condensing components, as well as low-temperature cooling plates. While the device can be used for frost or dew of samples, its chambers could not provide a true natural frost environment because leaf temperature decreased too fast, which was inconsistent with temperature changes in the natural environment. Fuller (1998) [20] designed a freezing chamber that was based on radiative cooling. The chamber has a rectangular design with a radiative cooling plate at the top cooled to −40 to −45 °C using HFC coolants. The cooling plate acted as a cold source, while the inner walls of the chamber were also cooled to variable temperatures as low as −5 °C in order to prevent the chamber walls radiating to the plant material during testing. The testing results showed radiation frosts and subsequent frost damage to potatoes in the air temperature range of −4 °C to −5 °C, and this equipment is recommended for studies of frost damage to plants normally caused by episodic radiation frost events. However, instantaneous variation in plant temperature was far less when plants were frozen in the chamber compared to the field. Indeed, in the field there was considerable "thermal noise", which meant that the leaf exotherms could not be determined whilst in the chamber the "thermal noise" is absent. We design a solation unit to avoid the thermal noise, which isolates the temperature and humidity in the cold radiation source from the basic refrigeration unit, to decrease of the leaf temperature. Frederiks et al. (2012) [21] summarized recent research of a frost simulation device, pointing out the disadvantages of convection from the frost chamber and explaining the characteristics and problems of the radiation cooling box. However, this study was more descriptive, and the main technical parameters and procedure was not clearly discussed.

The objective of this study was to provide a detailed design of an artificial radiation frost chamber to further frost-related research as well as to mimic the frost-forming process in the chamber as close as possible to actual conditions. Therefore, micrometeorological data and frost formation processing in

an experimental tea field were monitored during a typical radiation frost night. The radiation frost chamber model and its main parameters, such as the heat load calculation, thermodynamics calculation, and cooling unit, were determined by conducting the computational fluid dynamics (CFD) simulation on the leaf-air temperature difference. A frost-forming experiment was conducted to evaluate the performance of the chamber.

## 2. Materials and Methods

### 2.1. Observation of the Natural Frost Formation Process

The experiments were conducted on three nights in January 26 to 27, February 4 to 5, and March 1 to 2, 2016 that had calm weather on the former and cloudless conditions on the latter two. The observation of the frost formation process was conducted from 16:00 pm to 6:00 am during the three nights. Experimental tea farm (latitude of 32°02′35″ N, longitude of 119°67′80″ E) was located at Danyang, Jiangsu Province, which was planted with five years old Maolu tea variety and it has a canopy height of approximately 1.5 m. The tea farm has an average altitude of 18.5 m in the lower reaches of Yangtze River, which is the main production area for the famed and high-quality green tea [6] (Luo, 2015). Air temperature ($T$) and air relative humidity ($RH$) at a height of 1.0 m was measured with the HC2S3 (Campbell Scientific, Logan, UT, USA), while wind speed ($u$) was measured by a 2D sonic wind sensor (Wind Sonic C1-L, Campbell Scientific, USA) at a height of 2.5 m. During early winter and early spring (December, February, and March), the lowest air temperature ($T_{min}$) was −6 °C, $RH$ was greater than 90% and $u$ was always less than 1.5 m·s$^{-1}$ during frost nights [22,23]. Leaf temperature ($T_l$) was collected by a leaf and bud temperature sensor (SF110, Apogee Instruments, Logan, UT, USA) at a collection interval of 1.0 min.

### 2.2. Main Structures of the Frost Chamber

The technical parts of frost chamber include the basic refrigeration unit, the cold radiation source, and the isolation unit. As shown in Figure 1, the plant samples are placed in the basic refrigeration unit and their radiation heat through the isolation unit is absorbed by the radiation refrigeration unit, which leads to a decrease of the leaf temperature. When the leaf and air temperature difference ($T_{differ}$) reduces to a certain value, the water vapor in the air will sublimate into frost on the leaf surface. The basic refrigeration unit provides a basic environment and heat transfer will occur between the samples and radiation heat source unit. The isolation unit isolates the temperature and humidity in the cold radiation source from the basic refrigeration unit, and it is also used to increase the thermal efficiency of cold source radiation.

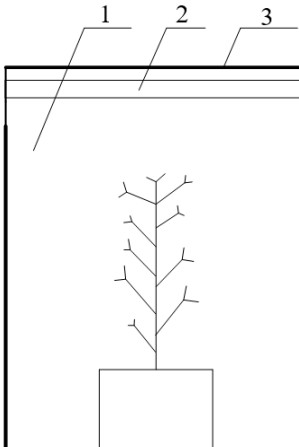

**Figure 1.** Technical parts of the frost chamber. 1. Basic refrigeration unit; 2. Isolation unit; and, 3. Cold radiation source.

As shown in Figure 2, the length, width, and height of the basic refrigeration unit is 800 mm, 400 mm, and 600 mm, respectively. It is composed of external and internal thermal insulation as well as a cooling plate. External thermal insulation minimizes the influence of hot air outside of the basic refrigeration unit; the cooling plate, located between the internal and external thermal insulation, is used to absorb the external heat load and it also adjusts the basic temperature to reduce the thermal load of radiation cooling.

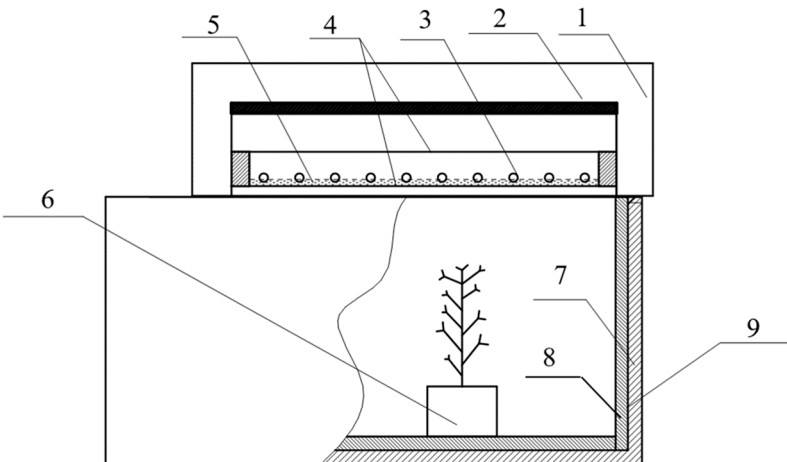

**Figure 2.** Structure of the frost chamber. 1. Thermal-protective coating; 2. Cold radiation source; 3. Heating wire; 4. High transmittance material; 5. Desiccant; 6. Plant sample; 7. External thermal insulation; 8. Internal thermal insulation; and, 9. Cooling plate.

The cold radiation source unit length, width, and height are 400 mm, 450 mm, and 150 mm, respectively, and it is fixed above the isolation unit and used to generate the low temperature. The isolation unit is fixed between the cold radiation source and basic refrigeration unit and it has a length, width, and height of 700 mm, 450 mm, and 600 mm, respectively; its outer surface isolation is coated with a thin film with high long wave transmittance to ensure enough radiation heat transfer between the sample and cold source. In addition, a heating wire is set up in the isolation unit to balance the cold air that is provided by the cold radiation source. Desiccant is fixed on either side of the isolation unit to prevent water vapor from condensing on the lower film.

## 2.3. Simulation of $T_{differ}$

A suitable $T_{differ}$ is the crucial factor for frost formation on the leaf. CFD simulation was conducted to determine the technical parameters after estimating the temperature of cold radiation source unit ($T_{radiation}$) and basic refrigeration cold source unit ($T_{basic}$), and the heating power of the external ($P_{ext}$) and internal thermal insulation ($P_{int}$). The optimization indexes were adjusted to the $T_{differ}$, and the meshing, boundary conditions, and numerical solution were also conducted for the simulation (Table 1).

**Table 1.** Aspects of the boundary conditions.

| Boundary Conditions | Cold Radiation Source | Inner Wall of the Basic Refrigeration Unit | Leaf | Heat | Film |
|---|---|---|---|---|---|
| Temperature (°C) | −30.0 | 0 | Coupling | | Coupling |
| Emissivity | 0.8 | 0.18 | 0.3 | 0.3 | 0.1 |
| Heat flux (W) (W/m²) | | | | 1000 | |
| Property of light transmission | Opaque | Opaque | Opaque | Opaque | Translucent |
| Thickness (m) | 0 | 0 | 0.002 | | 0.008 |

Three-dimensional (3D) drawing software (UG 11.0) was used to construct the 3D model of the frost chamber Figure 3a and it was subsequently imported to the grid software (ICEM 15.0). The hexahedral mesh division method was applied to the model to attain the advantages of high mesh quality and easy convergence, while the O-type meshing method was used around heating wire. The mesh boundary was encrypted with a total of 684,976 mesh cells and 646,017 nodes. The grid quality inspection showed a minimum determination coefficient of 0.5 and minimum aspect ratio of 16, which met the requirements of the numerical calculation, so the mesh was converted into an unstructured grid and then saved as a mesh file Figure 3b.

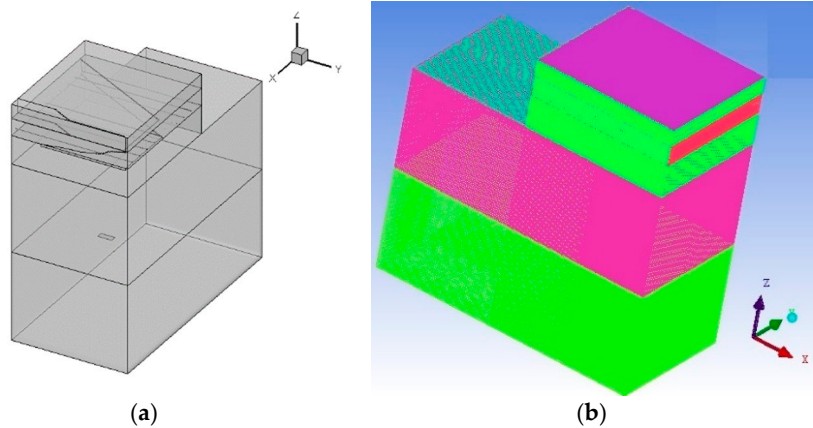

(**a**)                         (**b**)

**Figure 3.** Three-dimensional (3D) and Meshing models of the frost chamber, in the figure: (**a**) was the 3D model; and, (**b**) was the meshing model.

The detailed numerical simulation steps were showing as follows:

(1) import mesh file; conduct mesh check and smoothing; set proportional size; gravity acceleration was set to −9.81 m /s$^2$; and, pressure-based steady-state solution was selected.

(2) Energy equation and radiation heat transfer equation were added; the radiation model and DO model were chosen to calculate the semi−transparent medium and the flow equation with standard k-ε model.

(3) The gas was set as an ideal incompressible gas, and the absorption coefficient of the film was set as the following:

$$I_o = I_i e^{-\alpha x} \tag{1}$$

where, $I_o$ is the output energy, W; $I_i$ is the input energy, W; $x$ is the characteristic length, m; and, $\alpha$ is the absorption factor, m$^{-1}$. The transmittance of PE film is 80% and the thickness is 0.08 mm, he $\alpha$ is 2790 m$^{-1}$.

(4) The boundary condition setting is shown in Table 1. Heating power is expressed by heat flow density in Fluent, and the relationship between heat flow density and power is

$$\varphi = \frac{p}{\pi dl} \tag{2}$$

where, $\varphi$ is the heat flow density, W·m$^{-2}$; $P$ is the heating power, W; $d$ is the diameter of hot heating wire, m; $l$ is the axial length of heating wire, m. $P$, $d$ and $l$ is set to 40 W, 0.001 m and 0.07 m, respectively. So the $\varphi$ is 1000 W·m$^{-2}$.

(5) Monitoring value was set and the default residual convergence standard was chosen; leaf temperature, lower film temperature and upper film temperature were selected as monitoring items.

(6) Simple algorithm and standard pressure option and first−order upwind format were selected.

(7) After initializing the flow field, the number of iterations was set to 5000 before start the iteration. After the calculation of convergence, the second−order upwind format was used to continue the iteration until convergence.

### 2.4. Main Circuit Design for the Frost Chamber

The main control circuit design consists of a circuit breaker (QF), thermal relay (KM1, KM2, and KM3), fuse, time relay, radiative refrigeration compressor (RRC), basic refrigeration compressor (BRC), decompression valve (DV), 24 V DC battery, and heating wire. The control circuit includes an automation temperature controller (ATC1, ATC2, and ATC3), the time relay coil, and the switch relay coil; the automation temperature controller received signals that were collected by the temperature sensor for controlling the main circuit (Figure 4).

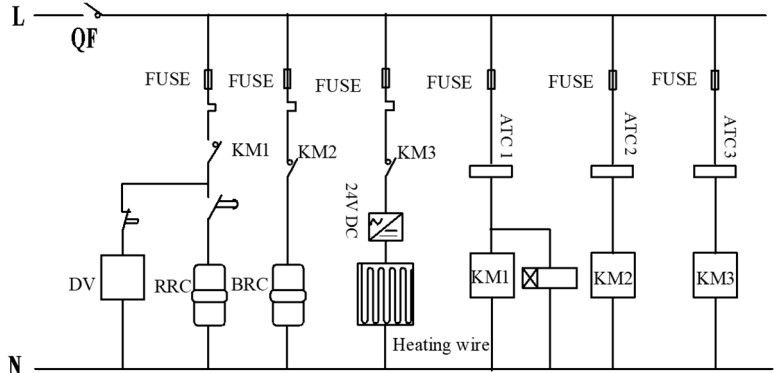

**Figure 4.** Circuit design of the frost chamber controller.

### 2.5. Performance Test of the Frost Chamber

#### 2.5.1. The Cooling Ability of the Cold Radiation Source Unit and Basic Refrigeration Unit

The sampled tea cultivar is Maolu, which was well maintained in the greenhouse that was located in the experimental tea field. The tea plants that were used for the experiment had healthy leaves and they were sampled on March 28, 2016, when the ambient temperature was 18.0 °C. Performance testing was conducted on the following day, March 29, 2016, which had the same ambient temperature as the day before. Table 2 shows the temperature setting of the cold radiation source unit ($T_{radiation}$) and basic refrigeration cold source unit ($T_{basic}$). In each treatment, only turned on the basic refrigeration unit was chosen as the control group.

**Table 2.** Temperature settings.

| $T_{radiation}$ (°C) | $T_{basic}$ (°C) |
|:---:|:---:|
| −40.0 | −5.0 |
| −20.0 | 0 |
| 0 | 5.0 |

#### 2.5.2. Temperature Control Scheme

In this study, $T_{radiation}$ and $T_{basic}$ were continuously maintained to realize the continuous decrease of $T_l$. The cold radiation source unit system was opened first to decrease the $T_l$, followed by opening the base refrigeration unit to absorb the heat load outside the box and reduce the heat load of the radiant refrigeration. As the heat load decreased, the cold radiation source could further reduce the $T_l$.

To determine the temperature control parameters, the change in $T_l$ under different $T_{radiation}$ was tested. The initial $T_l$ was set to 10.0 °C, $T_{radiation}$ was set to 7.5 °C, and $T_{radiation}$ was set to 0, 10.0, −20.0, −30.0, and −40.0 °C.

## 3. Results and Discussion

### 3.1. Observation Results

During the three tea field observation nights, $T_{differ}$ showed an obvious decrease from 16:00 to 18:00 (Figure 5), as the cooling rates ($V_{cool}$) were both 3.5 °C·h$^{-1}$ on January 26 and February 4, and that was 2.9 °C·h$^{-1}$ for March 1. There was no frost on the night of January 26 to 27, while only a minute ice crystal formed on the tea leaf at 19:00 on March 1. While the ice crystals gradually disappeared with the increase of $u$, subtle frost formations were observed on the leaf from 1:00 to 4:00 on March 2. At 18:00 of February 4, some tiny ice crystals formed on the tea leaf, which entirely covered the leaf by 20:00. The lowest leaf−air temperature difference ($T_{differ-min}$) was −1.0 °C on the night of January 26 to 27, as it altered between 0 °C and 1.0 °C due to the cloudy weather condition. The desublimation rate of the water vapor in the air ($V_{de}$) was lower than the sublimation rate of frost ($V_{sub}$), which did not result in any frost formations on the leaf. On the clear and calm night of February 4 to 5, $T_{differ-min}$ was −2.3 °C and it was kept stable at around −1.1 °C. When $V_{de}$ was higher than the $V_{sub}$, it resulted in frost formation on the leaf. $T_{differ-min}$ was −1.0 °C on the night of March 1 to 2, whereby the $V_{de}$ was similar to the $V_{sub}$, which resulted in no noticeable frost on the leaf.

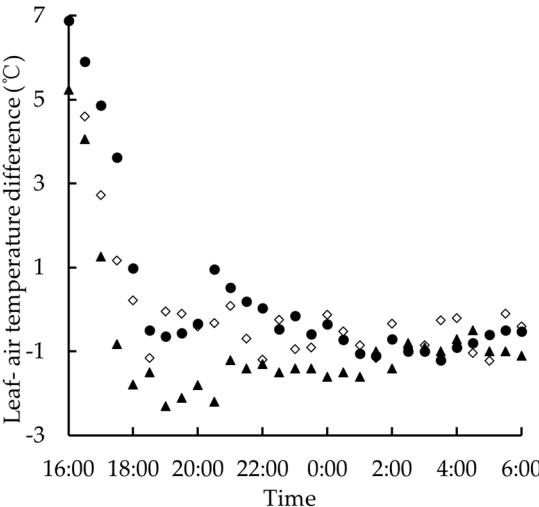

**Figure 5.** $T_{differ}$ in the three observation nights, where (▲), (●), (◇) is represented for the dates of February 4 to 5, March 1 to 2, and January 26 to 27, 2016, respectively.

### 3.2. Simulation Results

The simulation results showed that, when $T_l$ was −2.5 °C and cabinet air temperature was 0 °C, $T_{differ}$ was −2.5 °C, which did not meet the design requirement. Thus, further reduction of the temperature of the cold radiation source was needed. The refrigerating fluid used in the chamber was Freon and its lowest boiling point was −50.0 °C under normal temperature and pressure. While taking the load of the refrigeration compressor into consideration, $T_{radiation}$ was set to −40.0 °C. The power of the heating wire was set to 40.0 W based on the theoretical calculation, and it showed a condensation degree of 0 on the upper and bottom films of the isolation layer. As shown in Figures 6 and 7, $T_{differ}$ then reached −3.2 °C, meeting the design requirement.

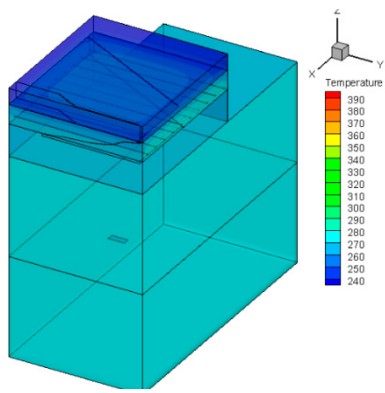

**Figure 6.** Temperature contour.

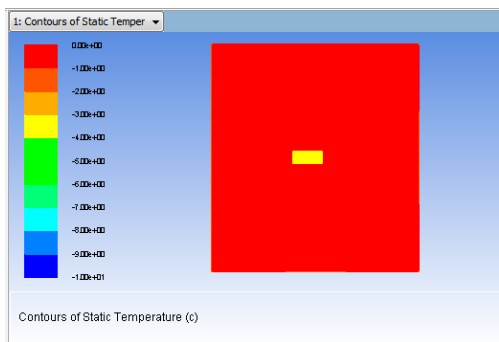

**Figure 7.** Leaf−air temperature difference.

### 3.3. Performance Experiment Results

#### 3.3.1. $T_l$ Variation

As $T_{basic}$ was set to −5.0 °C, $T_{radiation}$ was set to −40.0, −20.0, and 0 °C, the result showed that the decrease rate of $T_l$ was 14.0 °C·h$^{-1}$ at the first 0.5 h, which was same as the control group. This means that the cooling rate of $T_l$ was primarily affected by the basic refrigeration unit. After the combined action conducted over 3.0 h, $V_{cool}$ became higher than that of the control group, whereas $T_l$ was −3.8, −2.9, and −0.4 °C, respective to $T_{radiation}$ set at −40, −20, and 0 °C. Therefore, the cold radiation source further reduced $T_l$ when the influence from basic refrigeration unit tended to be stable (Figure 8a).

$T_{basic}$ was set to 0 °C and at the first 0.5 h, $V_{cool-leaf}$ was 12.0, 12.0, and 10.0 °C·h$^{-1}$, respective to $T_{radiation}$ set at −40.0 °C, −20.0 °C, and 0 °C. $V_{cool-leaf}$ of the control group was 7.0 °C·h$^{-1}$. After the combined action conducted over 3.0 h, $T_l$ was −0.9, 2.7, 2.4, and 4.3 °C. This showed that there was an enhanced effect of the cold radiation source unit on $T_l$ (Figure 8b).

$T_{basic}$ was set to 5 °C and at the first 0.5 h, $V_{cool-leaf}$ was 9.8, 7.8, and 4.0 °C·h$^{-1}$, respective to $T_{radiation}$ at −40.0, −20.0, and 0 °C. $V_{cool-leaf}$ of the control group was 2.6 °C·h$^{-1}$. After the combined action conducted over nearly 3.0 h, the $T_l$ was 2.3, 3.8, 7.0, and 7.8 °C, respectively. This showed a further enhanced effect of the cold radiation source unit on $T_l$ strengthened (Figure 8c).

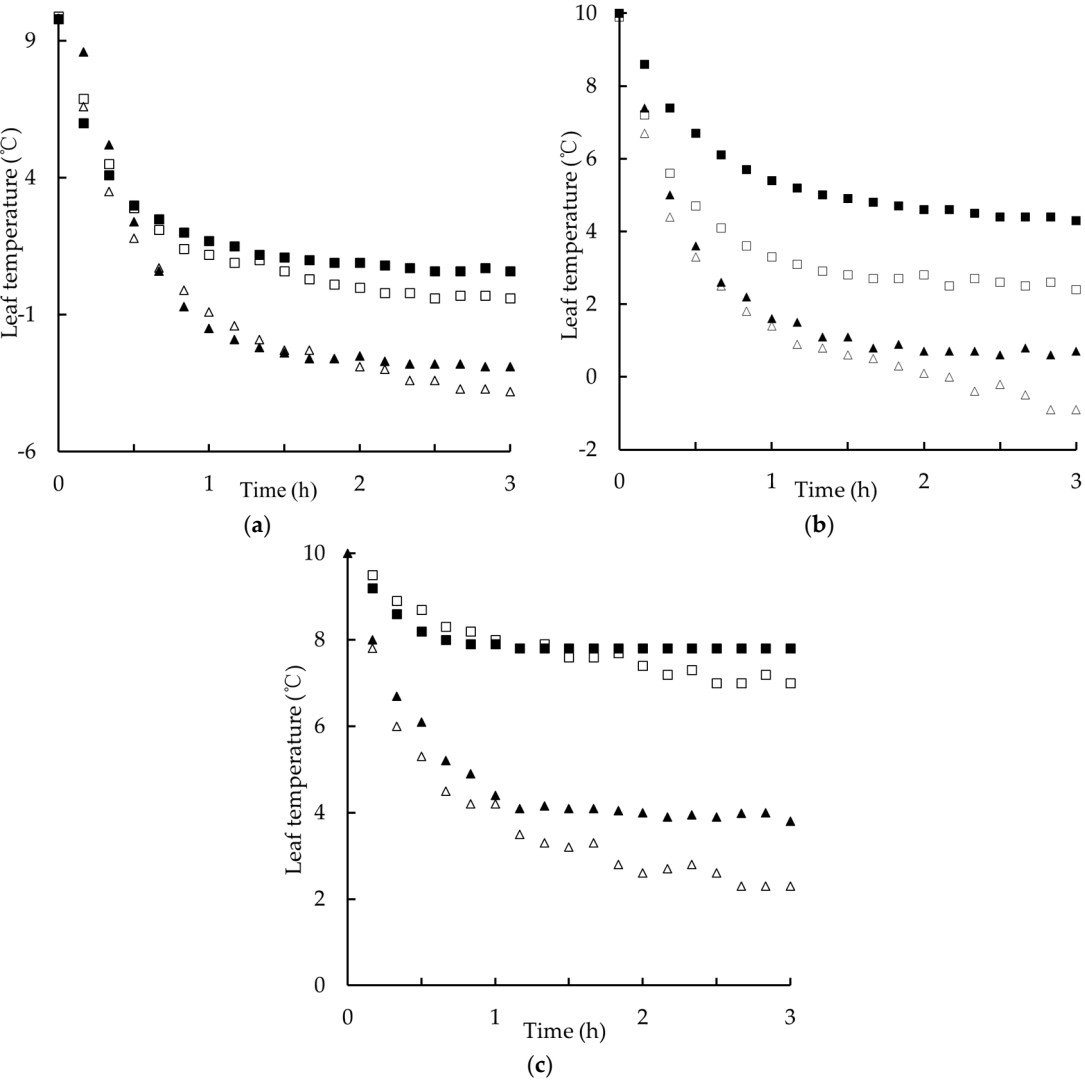

**Figure 8.** $T_{basic}$ was set to −5.0 °C (**a**); 0 °C (**b**); and, 5.0 °C (**c**) while the $T_{radiation}$ was set to −40 °C (△), −20 °C (▲), 0° C (□), and control (■), respectively.

### 3.3.2. $T_{differ}$ Variation

$T_l$ was influenced by the combined action of the cold radiation source and basic refrigeration units (Figure 9). $T_{differ}$ was used as an index to evaluate the cooling capacity of the cold radiation source. $T_{radiation}$ was set to −40.0 °C, and $T_{differ}$ was −4.4, −5.2, and −5.5 °C, relative to $T_{basic}$ at −5.0, 0, and 5 °C. Furthermore, $T_{radiation}$ was −20.0 °C, $T_{differ}$ was −3.5, −3.6, and −3.7 °C, respectively. For $T_{radiation}$ at 0 °C, $T_{differ}$ was −1.0, −1.1, and −0.8 °C, respectively. These results showed that different $T_{basic}$ had little influence on the cooling ability of the cold radiation source. $V_{cool-leaf}$ was similar to that of the natural environment under the individual effect of the cold radiation source within the first one hour. Following those results, $T_l$ slowly decreased.

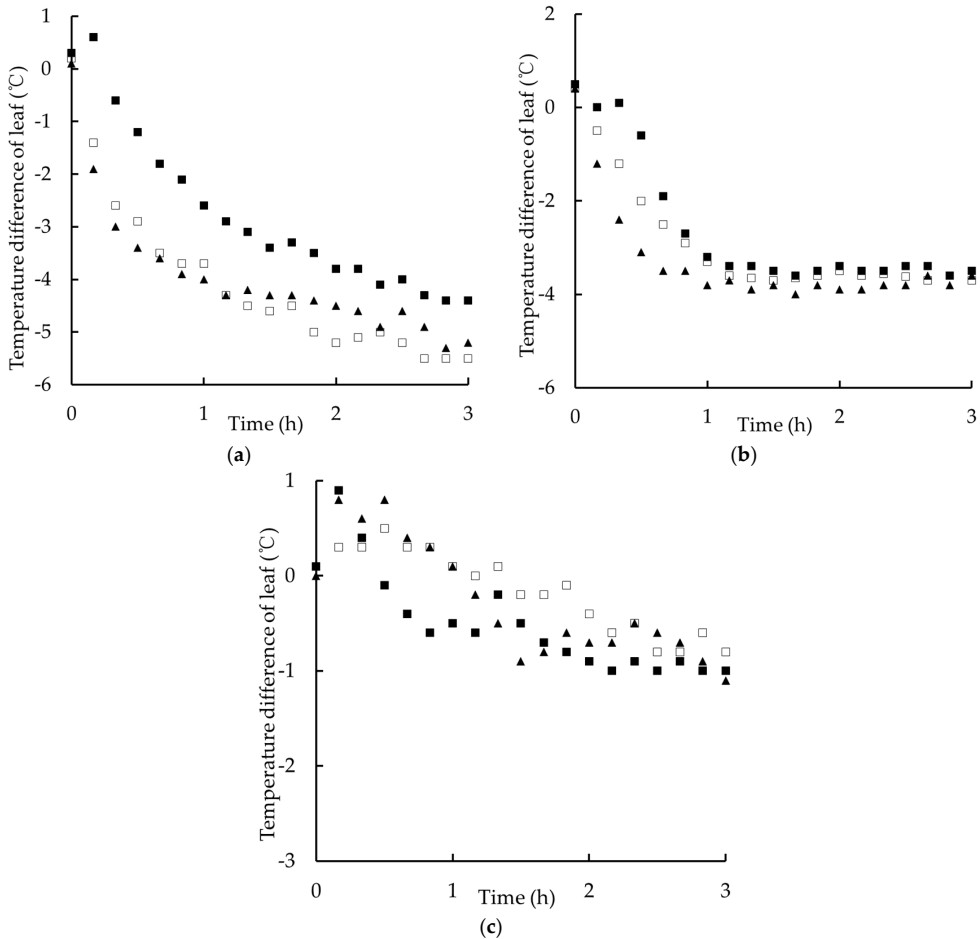

**Figure 9.** $T_{radiation}$ was set to −40 °C (**a**); −20 °C (**b**) and 0 °C (**c**), while the $T_{basic}$ was set to −5 °C (■),
0 °C (▲), 5 °C (□), respectively.

### 3.3.3. Modification of Initial $V_{cool}$

$V_{cool}$ in the chamber was dissimilar to the natural environment after the system had initially
functioned for 1.0 h (Figure 10). Thus, it was necessary to provide a new kind of temperature control
scheme to make sure that it was consistent with the natural conditions in order to make $T_l$ decrease
10.0 °C in the first 2.0 h.

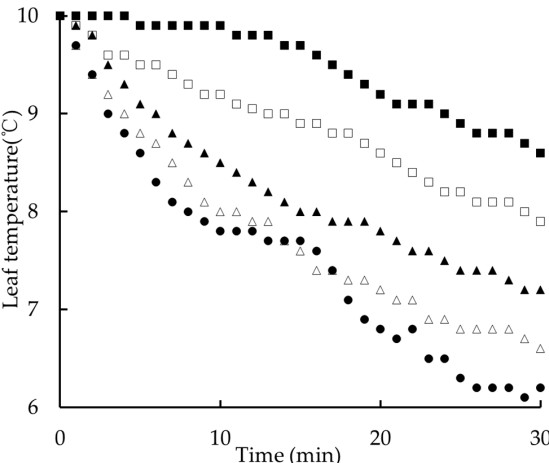

**Figure 10.** $T_l$ when the $T_{radiation}$ was set to 0 °C (■), −10 °C (□), −20 °C (▲), −30 °C (△), and −40 °C (●).

When $T_{radiation}$ was set to 0, 10.0, 20.0, 30.0, and 40.0 °C, $T_l$ decreased 1.4, 2.1, 2.8, 3.4, and 3.8 °C, respectively, within the first 0.5 h (Figure 10). Therefore, these cooling trends can be used to control $T_l$ and Table 3 shows the adjustment scheme. Firstly, $T_{basic}$ was set to 7.5 °C, and it rapidly decreased the $T_l$ to 10.0 °C. Meanwhile, $T_{radiation}$ was set to −10 °C and kept at that temperature for 0.5h, and then $T_{basic}$ was set to 5.4 °C. After that, $T_{radiation}$ was set to −20.0 °C and kept at that temperature for 0.5 h, and then $T_{basic}$ was set to 3.6 °C. Subsequently, $T_{radiation}$ was set at −30.0 °C and kept for 0.5 h, after which the $T_{basic}$ was set at −1.2 °C. Finally, $T_{radiation}$ was set at −40 °C and it was kept there for 0.5 h. As shown in Figure 11, the change in actual $T_l$ was consistent with that of an ideal one in which the natural cooling process for a frost night was realized.

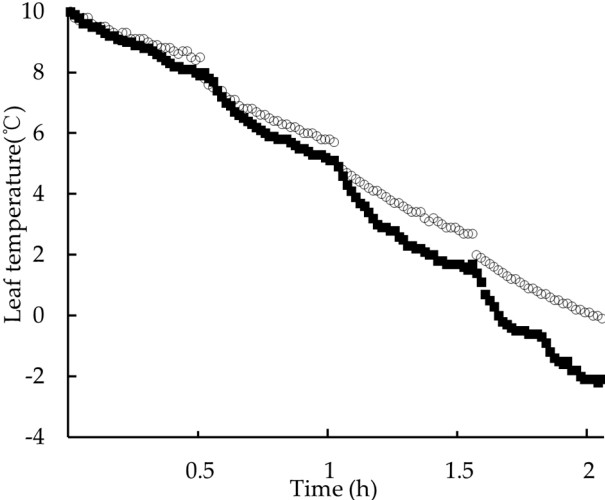

**Figure 11.** Comparison between the $T_l$ of actual (○) and ideal (■).

**Table 3.** $T_l$ adjustment scheme.

| $T_{radiation}$ (°C) | $T_{basic}$ (°C) | Cooling Time (h) | Ideal $T_l$ (°C) |
|---|---|---|---|
| −10.0 | 7.5 | 0−0.5 | 7.9 |
| −20.0 | 5.4 | 0.5−1.0 | 5.1 |
| −30.0 | 2.6 | 1.0−1.5 | 1.7 |
| −40.0 | −1.2 | 1.5−2.0 | −2.1 |

## 4. Conclusions

In a typical radiation frost night, the desublimation rate of the water vapor in the air must be greater than the sublimation rate of frost, which could cause frost formations on the leaf. Observation results showed that greatest $T_{differ}$ was −2.3 °C in the natural frost night. $T_{differ}$ should be greater than −3.2 °C to meet the design requirement. The performance testing results showed that the leaf temperature slowly declined after a rapid decrease. Additionally, the processes of frost formation on the surfaces of tea leaf samples in this chamber is similar to the naturally frost night. The change in actual $T_l$ was consistent with that of the natural conditions in frost night.

Used routinely for research aimed at understanding the physiology and mechanisms of frost injury to tea plants, the chamber was running for over 10 months and proved to be reliable and easy to use at normal temperature (around −10 to 25 °C). If the chamber works in the summer (ambient temperature higher than 25 °C), the frost formation is not obvious. We think it is caused by the bad airtightness between radiation cold source, thermal-protective coating, and basic refrigeration unit. We will keep on improving the chamber to make it potentially used in all kinds of ambient temperature. It could be applied in plant physiology research where dew formation is required, or where energy

exchange is to be studied. The chamber should be further studied to improve the control over the amount of frost.

**Author Contributions:** Y.-Z.L. and Y.-G.H. conceived and designed the experiments. Y.-Z.L. and J.-T.T. performed the experiments. Y.-Z.L. analyzed the data and wrote the manuscript. H.S. and R.L.S. gave some significant comments to improve the quality and language of the manuscript.

**Funding:** This research was funded by Jiangsu Agriculture Science and Technology Innovation Fund (CX(16)1045), Key R&D Programs of Jiangsu Province (BE2016354), Project of Postgraduate Innovation of Jiangsu Province (KYCX17-1788) and China Postdoctoral Science Foundation (2016M600376).

**Acknowledgments:** The authors are grateful for the APC finical support from Priority Academic Program Development of Jiangsu Higher Education Institutions (2014-37). The first author is many thanks to the China Scholarship Council (201708320220) for providing 24 months scholarship for studying in UC Davis.

**Conflicts of Interest:** The authors declare no conflict of interest.

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
