# Peer review of "Artificial Radiation Frost Chamber for Frost Formation on Tea"

_applsci, doi:10.3390/app9224726_

Round 1
Reviewer 1 Report
Below you can find my comments about the paper
“Artificial radiation frost chamber for frost formation on tea”
-My main concern is what new is learned from this study. What more has been found experimentally and computationally? The authors must try to bring out this message.
- Please separate experiments from numerics. Please consider to present in different and well-defined sections all experimentally related methods and results, and in other sections the numerical methods and results. Input and outputs of each methodology must be clearly presented. Explain better why there is need to perform both experiments and numerical simulations.
- What is missing from this work is equations. Some details regarding boundary conditions are simply given without explaining at all. It is questionable whether the study as it is can be reproduced. No information is provided about the equations and boundary conditions that are solved using CFD. Numerical details and methods are totally absent. How the leaf is treated numerically? How was it discretised?
-Before presenting the results, please summarise what cases were examined and why, as well as the range of all relevant parameters.
- I don’t see any mesh in Figure 3. What the authors mean with CFD model? This figure seems to represent the model geometry. Please remove ANSYS logo from the figure.
Author Response
Dear reviewer,
Thank you for your letter and the reviewers’ comments concerning our manuscript entitled” “Artificial radiation frost chamber for frost formation on tea”. (ID: applsci-631722). Thank you for giving our opportunity to revise our manuscript. Those comments are all valuable and very helpful for improving our manuscript. We have studied the comments carefully and have made correction which we hope will meet your request and for approval.
The reviewer’s comments are marked in yellow and our responses are marked in green.
Best,
Yongzong Lu and Yongguang Hu.
Q1: My main concern is what new is learned from this study. What more has been found experimentally and computationally? The authors must try to bring out this message.
R: Thanks for your concern. The objective of this study was to evaluate an artificial radiation frost chamber based on the temperature difference between leaf and air dew point, which was designed for advanced frost-related research. In most frost studies, researchers normally used an incubator based on the convective heat transfer between plants and the air to decrease the environment temperature, however, this is inconsistent with the mechanism of a natural radiation frost event. So frost chamber is crucial for frost related research and there were rare studies present the detailed design of this kind of chamber due to our literature review in recent three decades. What we found in our experiment and CFD simulation is that the greatest temperature difference between leaf and air dew point was -2.3°C, and the desublimation cooling rate of the air vapor was greater than sublimation, which caused frost to easily form on the tea leaf.
Q2: Please separate experiments from numerics. Please consider to present in different and well-defined sections all experimentally related methods and results, and in other sections the numerical methods and results. Input and outputs of each methodology must be clearly presented. Explain better why there is need to perform both experiments and numerical simulations.
R: Thanks for your kind comments for revising and improving our manuscript. The radiation frost chamber model and main parameters were determined by theoretical calculations and computational fluid dynamics (CFD) simulation. That is why we conducted the CFD simulation. A frost-forming experiment was conducted to evaluate the performance of the frost chamber. The object of this research is to present a detailed design of a kind of radiation frost chamber. So we both performed the experiments and numerical simulations.
For the results sections, we defined them as 3.1 Observation results, 3.2 Simulation Results and 3.3 Performance experiment results, which we think these are well defined.
The relevant corrections have been made in revised version in the manuscript.
Q3: What is missing from this work is equations. Some details regarding boundary conditions are simply given without explaining at all. It is questionable whether the study as it is can be reproduced. No information is provided about the equations and boundary conditions that are solved using CFD. Numerical details and methods are totally absent. How the leaf is treated numerically? How was it discretised?
R: Thanks for your kind comments for revising and improving our manuscript. We added the details regarding boundary conditions and numerical details in the section 2.3. Hope it can be well presented the CFD simulation details.
“ The detailed numerical simulation steps were showing as follows:
(1) import mesh file; conduct mesh check and smoothing; set proportional size; gravity acceleration was set to -9.81m /s2 and pressure-based steady-state solution was selected.
(2) energy equation and radiation heat transfer equation were added; the radiation model and DO model were chosen to calculate the semi-transparent medium and the flow equation with standard k-ϵ model.
(3) the gas was set as an ideal incompressible gas, and the absorption coefficient of the film was set as following:
(1)
where, Io is the output energy, W; Ii is the input energy, W; x is the characteristic length, m; α is the absorption factor, m-1. The transmittance of PE film is 80% and the thickness is 0.08mm, he α is 2790 m-1.
(4) boundary condition setting is shown in table 1. Heating power is expressed by heat flow density in Fluent, and the relationship between heat flow density and power is
(2)
where, φ is the heat flow density, W·m-2; P is the heating power, W; d is the diameter of hot heating wire, m; l is the axial length of heating wire, m. P, d and l is set to 40 W, 0.001 m and 0.07 m, respectively. So the φ is 1000 W·m-2.
(5) monitoring value was set and the default residual convergence standard was chosen; leaf temperature, lower film temperature and upper film temperature were selected as monitoring items.
(6) simple algorithm and standard pressure option and first-order upwind format were slected.
(7) after initializing the flow field, the number of iterations was set to 5000 before start the iteration. After the calculation of convergence, the second-order upwind format was used to continue the iteration until convergence.”
The relevant corrections have been made in revised version in the manuscript.
Q4: Before presenting the results, please summarise what cases were examined and why, as well as the range of all relevant parameters.
R: Thanks for your kind comments for revising and improving our manuscript. For the CFD simulation parts, a suitable Tdiffer is the crucial factor for frost formation on the leaf. So CFD simulation was conducted to determine the technical parameters after estimating the temperature of cold radiation source unit (Tradiation) and basic refrigeration cold source unit (Tbasic), and the heating power of the external (Pext) and internal thermal insulation (Pint). The optimization indexes were adjusted to the Tdiffer, and the meshing, boundary conditions and numerical solution were also conducted for the simulation.
For the performance experiment parts, Tradiation and Tbasic was maintained continuously to realize the continuous decrease of Tl. The cold radiation source unit system was opened first to decrease the Tl, followed by opening the base refrigeration unit to absorb the heat load outside the box and reduce the heat load of the radiant refrigeration. As the heat load decreased, the cold radiation source could further reduce the Tl. To determine the temperature control parameters, the change in Tl under different Tradiation was tested. The initial Tl was set to 10.0°C, Tradiation was set to 7.5°C and Tradiation was set to 0, 10.0, -20.0, -30.0 and -40.0°C.
The relevant corrections have been made in revised version in the manuscript.
Q5: I don’t see any mesh in Figure 3. What the authors mean with CFD model? This figure seems to represent the model geometry. Please remove ANSYS logo from the figure.
R: Thanks for your kind comments for revising and improving our manuscript. Sorry for the poor quality of the figure 3 which makes you did not clearly identify the mesh. We removed the ANSYS logo. We also changed the name of the figure 3b into 3-D model geometry.
The relevant corrections have been made in revised version in the manuscript.

Reviewer 2 Report
Line 22-24, The greatest temperature difference between leaf and air dew point was observed, but the prevention procedure from frost did not mentioned neither in the abstract nor the conclusion. It's suggested to be proposed by the authors.
Line 266, The authors mentioned the cooling capacity index of the cold radiation source, the Tdiffer , should be greater than -3.2°C. But the conclusion did not exactly point out the relationship between the index and the control methods from the amount of frost.
Abstract and conclusion description did not well match, it's suggested to be modified before acceptance.
Author Response
Dear reviewer,
Thank you for your letter and the reviewers’ comments concerning our manuscript entitled” “Artificial radiation frost chamber for frost formation on tea”. (ID: applsci-631722). Thank you for giving our opportunity to revise our manuscript. Those comments are all valuable and very helpful for improving our manuscript. We have studied the comments carefully and have made correction which we hope will meet your request and for approval.
The reviewer’s comments are marked in yellow and our responses are marked in green.
Best,
Yongzong Lu and Yongguang Hu.
Q1: Line 22-24, The greatest temperature difference between leaf and air dew point was observed, but the prevention procedure from frost did not mentioned neither in the abstract nor the conclusion. It's suggested to be proposed by the authors.
R: Thanks for your kind comments for revising and improving our manuscript. Sorry for making you confused.
We revised the abstract and conclusion and hope it can reach your requirements.
The relevant corrections have been made in revised version in the manuscript.
Q2: Line 266, The authors mentioned the cooling capacity index of the cold radiation source, the Tdiffer , should be greater than -3.2°C. But the conclusion did not exactly point out the relationship between the index and the control methods from the amount of frost.
R: Thanks for your kind comments for revising and improving our manuscript. Because our mistake on editing the manuscript, we lost all the minus when we described the Tdiffer in this paragraph. Sorry for making you confused. We revised the section 3.3.2 as following:
“Tl was influenced by the combined action of the cold radiation source and basic refrigeration units (Figure 9). Tdiffer was used as an index to evaluate the cooling capacity of the cold radiation source. Tradiation was set to -40.0°C, and the Tdiffer was -4.4, -5.2 and -5.5°C, relative to the Tbasic at -5.0, 0 and 5°C. Furthermore, Tradiation was -20.0°C, Tdiffer was -3.5, -3.6 and -3.7°C, respectively. For the Tradiation at 0°C, the Tdiffer was -1.0, -1.1 and -0.8°C, respectively. These results showed that different Tbasic had little influence on the cooling ability of the cold radiation source. The Vcool-leaf was similar to that of the natural environment under the individual effect of the cold radiation source within the first one hour. Following those results, the Tl decreased slowly.”
The relevant corrections have been made in revised version in the manuscript.
Q3: Abstract and conclusion description did not well match, it's suggested to be modified before acceptance.
R: Thanks for your kind comments for revising and improving our manuscript. We modified the conclusion and abstract as following:
Conclusion:
“In a typical radiation frost night, the desublimation rate of the water vapor in the air must be greater than the sublimation rate of frost which could cause frost formations on the leaf. Observation results showed that greatest Tdiffer was -2.3°C in the natural frost night. To meet the design requirement, the Tdiffer should be greater than -3.2°C. Performance testing results showed that leaf temperature slowly declined after a rapid decrease. And the processes of frost formation on the surfaces of tea leaf samples in this chamber is similar to the naturally frost night. The change in actual Tl was consistent with that of the ideal one in which the natural cooling process in frost night was realized.
Used routinely for research aimed at understanding the physiology and mechanisms of frost injury to tea plants, the chamber has been in operation for over 10 months and found to be both reliable and easy to maintain in the normal ambient temperature (around -10 to 25 °C). If the chamber works in the summer (ambient temperature higher than 25 °C), the frost formation is not obvious. We think it is caused by the bad airtightness between radiation cold source, thermal-protective coating and basic refrigeration unit. We will keep on improving the chamber to make it could be used in all kinds of ambient temperature. It could be applied in plant physiology research where dew formation is required, or where energy exchange is to be studied. The chamber should be further studied to improve control the amount of frost.”
Abstract:
“The Yangtze River region is the main production area for famous, high-quality tea in China. Radiation frost occurs frequently in this region, especially in the early spring during calm and clear nights, and it causes substantial damage to crops leading to huge economic losses for tea growers. The formation of frost is difficult to control experimentally due to the complexity and variability of the agro-micrometeorological environment. The objective of this study was to evaluate an artificial radiation frost chamber based on the temperature difference between leaf and air dew point, which was designed for advanced frost-related research. To mimic declining temperatures that are consistent with nature, micro-meteorological data and the frost formation process were monitored in an experimental tea field during typical radiation frost nights. The radiation frost chamber model and main parameters were determined by theoretical calculations and computational fluid dynamics (CFD) simulation. A frost-forming experiment was conducted to evaluate the performance of the frost chamber. Observation results showed that greatest temperature difference between leaf and air dew point (Tdiffer)was -2.3°C. Simulation results showed that the desublimation cooling rate of the air vapor was greater than sublimation, and the Tdiffer should be greater than -3.2°C, which could cause frost easily form on the leaf. Performance testing results showed that leaf temperature slowly declined after a rapid decrease, which is similar to the natural condition, resulting in noticeable frost formation on the leaf.”
The relevant corrections have been made in revised version in the manuscript.
